# Recombinant Bovine Growth Hormone-Induced Metabolic Remodelling Enhances Growth of Gilthead Sea-Bream (*Sparus aurata*): Insights from Stable Isotopes Composition and Proteomics

**DOI:** 10.3390/ijms222313107

**Published:** 2021-12-03

**Authors:** Josefina Blasco, Emilio J. Vélez, Miquel Perelló-Amorós, Sheida Azizi, Encarnación Capilla, Jaume Fernández-Borràs, Joaquim Gutiérrez

**Affiliations:** Department of Cell Biology, Physiology and Immunology, Faculty of Biology, University of Barcelona, 08028 Barcelona, Spain; emilio-jose.velez-velazquez@inrae.fr (E.J.V.); miquelperelloamoros@gmail.com (M.P.-A.); sheidaazizi@ub.edu (S.A.); ecapilla@ub.edu (E.C.); jaume.fernandez@ub.edu (J.F.-B.); jgutierrez@ub.edu (J.G.)

**Keywords:** metabolism, cytochrome-c-oxidase, citrate synthase, redox enzymes, δ^13^C/δ^15^N, white muscle proteome

## Abstract

Growth hormone and insulin-like growth factors (GH/IGF axis) regulate somatic growth in mammals and fish, although their action on metabolism is not fully understood in the latter. An intraperitoneal injection of extended-release recombinant bovine growth hormone (rbGH, Posilac^®^) was used in gilthead sea bream fingerlings and juveniles to analyse the metabolic response of liver and red and white muscles by enzymatic, isotopic and proteomic analyses. GH-induced lipolysis and glycogenolysis were reflected in liver composition, and metabolic and redox enzymes reported higher lipid use and lower protein oxidation. In white and red muscle reserves, rBGH increased glycogen while reducing lipid. The isotopic analysis of muscles showed a decrease in the recycling of proteins and a greater recycling of lipids and glycogen in the rBGH groups, which favoured a protein sparing effect. The protein synthesis capacity (RNA/protein) of white muscle increased, while cytochrome-c-oxidase (COX) protein expression decreased in rBGH group. Proteomic analysis of white muscle revealed only downregulation of 8 proteins, related to carbohydrate metabolic processes. The global results corroborated that GH acted by saving dietary proteins for muscle growth mainly by promoting the use of lipids as energy in the muscles of the gilthead sea bream. There was a fuel switch from carbohydrates to lipids with compensatory changes in antioxidant pathways that overall resulted in enhanced somatic growth.

## 1. Introduction

The possibility of obtaining fish of greater size and quality in shorter times, inducing their growth from the first stages of their life, represents a challenge in the improvement of aquaculture. Growth hormone (GH) is the main endocrine inducer of fish growth; therefore, its manipulation under controlled rearing conditions can be considered as a rapid approach to explore the maximum inherent growth capacity of a given species that could potentially be recruited for aquaculture purposes, through long-term domestication and selective breeding programs [1]. One of the most important aspects in the knowledge of GH in fish was the development of a GH transgenic salmon [2]. These authors demonstrated, by using recombinant bovine GH (rBGH, Posilac©), that the transgenesis resulted in a certain level of saturation for growth, as rBGH administration had a lower effect enhancing growth in transgenic animals compared to their corresponding control group.

The GH/insulin-like growth factor (IGF) system (GH/IGF) is the main regulator of somatic growth in vertebrates, and it has been extensively studied, also in fish [3,4,5,6,7,8,9,10,11,12]. Thus, the number of receptors for both hormones in different tissues and the activation of signalling pathways through them have been investigated in different species [13,14], including gilthead sea bream [10,11,12]. GH administration increases growth rates in fish [8]. Moreover, GH is a pleiotropic hormone involved in several vital processes in vertebrates, such as nutrition, metabolism, reproduction, physical activity, neuroprotection, immunity, osmoregulation and even social behaviour [15]. The biological actions of GH as a major growth and metabolic modulator have been utilized in different applied fields including aquaculture [2,11,12]. All these reinforce the multidisciplinary interest on GH and the need for progressing in its knowledge across vertebrates.

The metabolic function of GH/IGF axis is noticeable as it can regulate the three major metabolites (sugars, lipids and proteins) and thus significantly increase the animal growth rate, promote the growth of muscles and bones and decrease the fat content [16,17]. As in mammals [18], GH has a dual role in fish, since it exerts a lipolytic effect on target tissues such as liver and adipose tissue in vivo [12,19,20] and in vitro [12,21,22], and at the same time, it has a hyperglycaemic effect by mobilizing liver glycogen [23,24,25,26]. GH and IGF-I treatments, both singly and in combination, significantly increased the glycogen content of hepatocytes isolated from fed Artic char [27]. Thereby, in vivo GH administration suppressed the potential for glycolysis but stimulated gluconeogenic and glycogenolytic pathways. The mobilization of energy substrates, lipids and glycogen enables protein savings leading to greater growth. Thus, white muscle protein accretion has been observed in rainbow trout after GH injection [28,29], and the anabolic effects of GH on protein turnover have been reflected as increased oxygen consumption [30]. However, most of the GH effects in carbohydrate metabolism have been analysed on the liver tissue, and information on the GH-induced consequences in muscle is scarce. Therefore, a holistic view using proteomics could provide valuable information on the metabolic pathways that may have been modified after an exogenous GH treatment. This tool has made possible to demonstrate that a similar rapid growth phenotype can be achieved via selective breeding or by GH transgenic fish, although the proteomic profile between the two revealed differences in the metabolic pathways involved [31].

Stable isotope analysis (^13^C and ^15^N) in representative growth tissues, such as muscle, can offer valuable information about reserves turnover [32], because enzymes involved in catabolic processes, such as decarboxylation and deamination, show a preference for light isotopes [33], causing tissues to become enriched with heavier isotopes (e.g., ^13^C and ^15^N). This factor of discrimination is known as fractionation. Nitrogen fractionation (Δ^15^N) is a good marker of protein balance, because it reflects protein turnover and retention efficiency [32,34]. In previous studies, we have evaluated the use of stable isotopes as indicators of feeding balance for assessing the optimal nutritional conditions of growing fish [34,35,36,37]. In the case of gilthead sea bream, we have observed an inverse relationship between dietary protein content and muscle nitrogen fractionation and that this fractionation is also inversely related to the specific growth rate (SGR) [37]. Therefore, we hypothesized that the set of modifications and metabolic changes induced by GH administration should be reflected in the isotopic composition of the fish tissues.

In the present study, we analysed the effects of rBGH treatment in both fingerlings and juveniles of gilthead sea bream, focusing the attention to the metabolic responses of liver and muscle tissues. Using a variety of techniques, we aimed to provide a valuable knowledge of the adaptation of those tissues to maintain/prolong GH administration. Specifically, the recycling of the reserves was studied in depth by interpreting the isotopic composition changes, in addition to the proximal composition and the key enzymes activity of the intermediate and energy metabolism, as well as the redox balance. From a holistic perspective, the differences that occur in the muscle proteome are shown together with analysis of the global effects on specific metabolic pathways. This work complements our previous studies on the effects of rBGH on myogenic and lipolytic factors and the GH/IGF axis in gilthead sea bream [11,12]. Although previous research in fish shows that rBGH is a good inductor of growth, the current legislations and the consumers’ criteria are not in line with the idea of using rBGH in fish. The purpose of this study was never to apply this to fish, but to know their potential to grow to later on find approaches or natural conditions where the secretion of GH and/or IGF-I is enhanced.

## 2. Results

### 2.1. Growth Performance, Proximal and Isotopic Composition

The effects of rBGH injection on somatic growth parameters and feed conversion rate on gilthead sea bream fingerlings and juveniles are shown in Table 1. The final body weight was significantly increased in both age groups injected with rBGH (*p* < 0.05, *p* < 0.01, respectively) compared to their corresponding control fish, which resulted in a significant increase of SGR in both groups as well (*p* < 0.05, *p* < 0.001). The condition factor (CF) was modified neither in fingerlings nor in juveniles. The hepatosomatic index (HSI) was significantly reduced in both experimental groups (*p* < 0.01), but no differences were observed in mesenteric fat index (MFI) of rBGH-injected groups compared to their corresponding control groups. The musculosmatic index (MSI) obtained in juveniles did not show significant differences. The food conversion ratio (FCR) decreased in all rBGH-injected fish compared to control groups, but it was significant only in the case of juveniles.

The proximal composition, RNA, and DNA concentrations of white muscle after rBGH injection on fingerlings and juveniles are presented in Table 2. A significant decrease of lipids was observed at both ages/sizes in the experimental fish (*p* < 0.001) compared to their respective control groups. The rBGH group of juveniles practically doubled (*p* < 0.01) the glycogen reserves in white muscle and showed significantly increased RNA concentration (*p* < 0.001), which resulted in an increase of the RNA/DNA ratio in this group (*p* < 0.01). Notwithstanding, these changes in white muscle glycogen and RNA were not observed in fingerlings.

The proximal composition in red muscle and liver was also analysed in juveniles (Table 3). The decrease of lipid concentration was also observed in liver and red muscle of rBGH group, but the differences were only significant in liver (*p* < 0.05). As in white muscle, the red muscle glycogen was significantly increased in the rBGH-injected fish (*p* < 0.01), but on the contrary in liver this reserve was decreased (*p* < 0.05) compared to the controls.

The isotopic composition of white muscle and its reserves were significantly modified by rBGH administration in both fingerlings and juveniles (Table 4). δ^13^ C values of muscle, lipid and glycogen reserves were significantly increased in rBGH-treated groups in fingerlings and juveniles, revealing turnover differences of these reserves between experimental and control groups. Furthermore, δ^15^N of muscle and of its protein was significantly reduced in both fingerlings (*p* < 0.05) and juveniles (*p* < 0.01), and as a consequence, the nitrogen fractionation was significantly reduced, indicating a better use of the protein diet in rBGH groups.

### 2.2. Enzyme Activities and Protein Expression

The activities of the main antioxidants enzymes, catalase (CAT), superoxide dismutase (SOD) and two glutathione forms, glutathione peroxidase and glutathione reductase (GPx and GR) in liver are showed in Figure 1. The enzyme activity of CAT was significantly decreased in rBGH group (*p* < 0.01), meanwhile SOD showed a tendency to increase in this group compared to the control fish. GR significantly increased (*p* < 0.05) and GPx decreased (*p* < 0.05) in rBGH-injected group compared to control group.

In liver, the enzyme activity of citrate synthase (CS) significantly increased (*p* < 0.05) and that of alanine transaminase (ALAT) significantly decreased (*p* < 0.01) in rBGH-injected juveniles compared to the control group, but no changes were observed in hydroxyacyl CoA dehydrogenase (HOAD), lactate dehydrogenase (LDH) or aspartate transaminase (ASAT) (Table 3). The enzyme activities of CS and cytochrome-c-oxidase (COX), the two energy metabolism enzymes, were not modified by rBGH injection in the white muscle of fingerlings and juveniles (Table 2) nor in the red muscle of juveniles (Table 3). When the protein expression of these enzymes was analysed in both muscles of juvenile fish, only a significant lower protein expression of COX (*p* < 0.001) was observed in white muscle (Figure 2). In red muscle of the rBGH group, a significant lower expression of regulatory factor PGC1-alpha (*p* < 0.05) was also found (Figure 2).

### 2.3. White Muscle Proteome

When analysing the proteome, the number of spots detected in the white muscle of all samples of gilthead sea bream juveniles ranged between 1029 for rBGH group and 1025 for control group. Three-hundred candidate spots selected for mass spectrometry (MS) analysis presented a 2-fold change or an intensity exceeding 0.1% (relative to total intensity). Of these, 43 protein spots showed significant differences (*t*-test, *p* < 0.05) between the rBGH and control groups, being 25 down-regulated and 18 up-regulated in response to rBGH injection. The MS characterization of these proteins, followed by MASCOT database searches, resulted in 7 non-detected proteins (corresponding to trypsin and keratin residues) and 36 identified protein spots (Table 5), which corresponded to 13 protein sequences that have been already reported in teleost species. The results of Gene Ontology (GO) enrichment analysis indicated that most of these proteins (8) were grouped in a single cluster, “carbohydrate metabolic process” (GO: 00005975) (Figure 3A), and most of them were down-regulated the in white muscle of the rBGH-treated group, as it is shown in the heat map of the significantly different proteins presented in Figure 3B.

## 3. Discussion

This study is a continuation of the research conducted previously in our group to investigate the effect of rBGH on the GH/IGF axis and the regulation of myogenesis and osteogenesis in fingerlings [11] and juveniles [12] of gilthead sea bream. In the present work, the final body weight, SGR and somatic indexes of fingerlings are the same as those reported in Vélez et al. (2018) [11], but in juveniles, they correspond to samplings performed 9 weeks after injection, while in Vélez et al. (2019) [12], these growth parameters were obtained at 6 and 12 weeks after injection. The time points used, 6 and 9 weeks, were selected taking into account previous studies, as discussed in Vélez et al., (2019) [12]. In both experiments, we found an improvement of somatic growth that is in accordance with the observations of other authors made from fish species injected with rBGH [2,8,38,39]. Interestingly, we did not observe changes in CF and MSI, suggesting a harmonious and proportional growth of individuals, in contrast to what it was observed in coho salmon under similar experimental conditions [40]. Notwithstanding, the dose of GH used, the species and the fish age may determine different effects that would explain the variety of results among studies. In fact, we were able to observe decreased CF at 6 weeks after administration of rBGH [12], which subsequently disappeared, as we have observed in the 9-week results, and that was maintained up to week 12. Moreover, the stimulation of growth determined a significant reduction of FCR in rBGH-treated juveniles, suggesting an improvement of the food conversion efficiency, as other authors previously reported in other species [41,42].

Although the decrease on mesenteric fat (MFI) and red muscle lipid concentration on rBGH fish was not significant, the clear effects in the white muscle of both fingerlings and juveniles and in the liver of juveniles after rBGH injection verified the lipolytic effect of this hormone, in agreement with previous observations in other fish species [19,43,44,45]. The lipolytic effect of GH has been widely described, but few studies in fish have determined the reduction of the lipid reserves in target tissues. In this sense, the rBGH treatment induced a decrease in the subcutaneous fat layer and in plasma triglycerides levels of gilthead sea bream fingerlings [11], suggesting an enhancement of lipolytic activity. Moreover, increased release of glycerol was found in isolated hepatocytes, adipocytes and primary cultured myocytes extracted from rBGH-injected juveniles [12]. This key metabolic action of GH was also observed in isolated adipocytes incubated with homologous GH [46], and it is at least partly mediated via the hormone-sensitive lipase (HSL) activity [46,47,48]. In spite of the lipolytic role of rBGH, the activity of the liver HOAD, a fatty acid β-oxidation key enzyme, was not affected at 9 weeks post-injection in juveniles. Under the conditions of this study, the livers of the experimental animals had less lipids than those in the control group, which would indicate that the lipolytic effect of rBGH would have finished at 9 weeks or that other enzymes, such as HSL, would be activated [12], as it occurs in mammals [18]. This increased lypolitic activity may be viewed as a mean of switching substrate metabolism from glucose and protein utilization to lipid oxidation, as pointed out in mammals [18].

The changes of isotopic composition of the tissue reserves allowed us to deepen more in the use of these reserves, their turnover, and the resulting crosstalk among tissues such as liver and muscle. Thus, when the energy needs of gilthead sea bream vary as it happens during exercise [36,49], or when changes in the feeding conditions are induced [37], the tissues isotopic footprint is modified, revealing changes in the recycling of reserves and the allocation of C or N between diet and tissue reserves. Accordingly, the significant enrichment of the δ^13^C observed in white muscle of both rBGH-injected fish groups (i.e., fingerlings and juveniles) would be a consequence of the reduction in white muscle lipids, as we have demonstrated in other demanding energy conditions, such as swimming activity [36]. Moreover, the turnover of white muscle glycogen increased (i.e., increase in δ^13^C-glycogen) in juveniles, which would contribute to change the fingerprint of δ^13^C of muscle. The significant enrichment of the δ^13^C from the glycogen reserve, and the increase in concentration of this metabolite in both muscles would indicate a turnover modification of this reserve with a positive balance for the synthesis and/or decrease of its degradation as a consequence of rBGH injection. In this way, GH and IGF-I treatments, both singly or in combination, significantly increased the glycogen content of hepatocytes isolated from Arctic char [27], in agreement with that observed in mammals [18]. Thus, the present results showed that there is a shift in the preferential use of energy substrates in response to GH treatment also in gilthead sea bream, with a greater use of lipids and a reduction in carbohydrate utilization, as observed in mammals. In the latter, sustained glucose release is dependent on gluconeogenesis from amino acids, while increased fat utilization and diminished glucose utilization by GH action also decrease the need for protein breakdown [18].

The role that GH exerts on protein metabolism in mammals is well known; it modifies protein balance by stimulating the synthesis and reducing protein degradation [18]. Although net protein growth has also been evidenced in fish species [39], there is no indication about GH direct effects on the processes of synthesis and protein degradation. Our results clearly indicated a reduction in muscle protein fractionation in juveniles of gilthead sea bream due to the injection of rBGH, which is interpreted as a positive balance on protein growth that in turn can be associated with higher synthesis or with reduced degradation. In this sense, the reduction of hepatic ALAT in the group injected with rBGH would be another evidence of the reduction in protein recycling in this group, in agreement with that observed in other teleost injected with GH [27] or in in vitro treated hepatocytes [22]. The observed decrease in nitrogen fractionation, as a result of decreased δ^15^N, a reliable indicator of nutritional status in fish [37], implies a reduction in protein turnover and a protein sparing as we observed previously in the same species submitted to changes in diet or swimming activity [35,36,37,49]. Moreover, the significant increase found in muscle RNA concentration in rBGH-injected juveniles also suggested that protein synthesis capacity was increased, as we observed in exercised gilthead sea bream [36,49]. In accordance with this result, the 20% increase in body weight in juveniles injected with rBGH, without changes in the percentage of muscle mass or protein content, clearly indicated that there has been a net increase in protein muscle in these fish compared to the control group. Our results are in agreement with those obtained in GH-injected rainbow trout, where the GH treatment altered catabolic and anabolic processes, producing a leaner fish with a higher protein content with respect to control fish [50].

Growth, protein synthesis and many components of fuel metabolism are ATP-requiring processes that may increase energetic demands on mitochondria, but GH effects on mitochondrial function are not fully established. Here, we have observed an increase of CS activity in the liver of rBGH-injected juveniles, in agreement with that observed in tilapias injected with GH [51]. However, this increase was not coupled to an increase of oxidative phosphorylation because COX activity was not modified. Then, changes in mitochondrial compartments between respiratory chain (inner membrane) and tricarboxylic acid (TCA) cycle enzymes (matrix compartment) may indicate a functional adaptation of mitochondria to adjust to tissue-specific demands in response to exogenous rBGH and to the availability of energy sources. Recent studies in humans have recorded increased mitochondrial oxidative capacity and expression of mRNAs that encode mitochondrial proteins after GH exposure alone or in combination with exercise [52,53]. GH action may therefore activate proteins in the β-oxidation or TCA cycles (e.g., HOAD and CS enzymes) or other components of the mitochondrial fuel delivery and oxidation machinery [52]. In parallel, a reduction of CAT and GPx activities was found in GH-transgenic salmon [54], which could suggest a decrease in the formation of H_2_O_2_. Moreover, the higher activity of GR (responsible for returning GSSG to its reduced and antioxidant active form GSH) would indicate that the GSH could be directly involved in the reduction of the ROS produced by the increased lipid metabolism or that the GSH works as a reducing factor of other antioxidant enzymes such as peroxiredoxins. Indeed, in Antarctic fish, it has been observed that these enzymes represent an important line of defence against the increase in the rate of H_2_O_2_ formation [55]. In contrast to liver, in both red and white muscle, the activity and protein expression of CS were not modified, but a decrease in protein expression of COX, maintaining the enzyme activity, was induced in the white muscle of rBGH fish. This response would indicate a decoupling between the Krebs cycle and oxidative phosphorylation, perhaps to spare energy in this condition. Interestingly, this reduction in COX activity was not observed in red muscle, where the expression of the regulatory factor PGC1α was reduced. It is known that this transcription factor is the master regulator of mitochondrial biogenesis in mammals [56,57], interacting with nuclear respiratory factors involved in codifying *OXPHOS* genes. However, information in fish about the role of PGC1α in these processes is scarce [58,59,60].

The proteomic approach of the present study in white muscle revealed how rBGH injection down-regulated only the expression of proteins grouped in a single cluster, the carbohydrate metabolic process. These results are in agreement with those observed recently in the domesticated aquaculture strain of coho salmon, selected for enhanced growth [31] and characterized by unique changes in abundance of carbohydrate-processing proteins in contrast to wild type animals. The authors indicated upregulation of several enzymes involved in glycolysis, glycogenolysis, gluconeogenesis and the pentose phosphate pathway but downregulation of several proteins involved in glucose breakdown and rapid energy generation as LDH. They concluded that during the domestication process, the remodelling of muscle energy metabolism permitted increased efficiency of energy generation to support the costs of protein turnover and growth. In our case, the high number of alpha-1,4 glycan phosphorylase protein spots indicated a high turnover of the glycogen reserve in muscle stimulated by rBGH but accompanied with a reduction in the long-term aerobic and anaerobic glycolytic pathway, possibly due to a shift in whole-body fuel utilization, reflecting reduced carbohydrate oxidation and enhanced fat oxidation as discussed above. This allowed muscle glycogen stores to be restored. Then, the effects of GH on carbohydrate metabolism are complicated both in the short and long term and may be indirectly linked via the antagonism of insulin action [61].

Moreover, Causey et al. [31] also observed downregulation of calcium-regulated proteins involved in muscle contraction regulation in domesticated coho salmon versus wild type, in agreement with our results where we observed downregulated tropomyosin. Whether these changes underlie differences in muscle composition and contractile properties remains to be demonstrated. This is the first proteomic analysis approaching the effects of rBGH in fish, and the results are in agreement with the metabolic changes indicated before.

In summary, the obtained results show how the metabolic machinery of the fish changes in the face of exogenous GH administration. Globally, a change in the use of metabolites with oxidative purposes is observed, i.e., a reduction in the use of carbohydrates, increased use of lipids and protein savings resulting in enhanced somatic growth of gilthead sea bream at both fingerling and juvenile stages. Moreover, in the same line, the isotopic approach has demonstrated a reduction of protein oxidative use in rBGH-treated fish, favouring an increase of protein deposition in muscle and, thus, fish growth.

## 4. Materials and Methods

### 4.1. Fish and Experimental Design

Two experiments with different fish size were designed.

#### 4.1.1. Gilthead Sea Bream Fingerlings

Two hundred fingerlings of gilthead sea bream (initial body weight 1.0 ± 0.05 g) were obtained from a commercial hatchery in the north of Spain and reared in the facilities of the Faculty of Biology at the University of Barcelona. Fish were distributed into 8 cages of 37 L; each two cages were put together in a larger thank (4 tanks of 200 L, each cage with 25 fish and thus, 50 fish/tank), as previously described [11]. Fish were maintained in a sea water recirculation system at 23 ± 1 °C and photoperiod of 15L:9D h. They were fed ad libitum (near 10% body weight/day) five times a day with a commercial diet (Gemma Diamond Skretting, Burgos, Spain). Two cages of each tank were randomly assigned to a treatment group; Control or rBGH. Fish were anesthetized (MS-222 0.08 g/L) and intraperitoneally injected with a single dose (4 mg·g^−1^ body weight) of rBGH (Posilac©, Elanco Animal Health, Eli Lilly and Company) diluted 1:4 with sesame oil (Sigma-Aldrich, Tres Cantos, Spain) or with the same volume of sesame oil for the controls. After 6 weeks, the final biomass of each cage was obtained (*n* = 4 for each experimental group). Twenty fish per cage (80 fish per condition) were anesthetized and sacrificed in order to obtain body weight and length and the livers and mesenteric fat to calculate both HSI and MFI, respectively. Moreover, SGR and CF were calculated for each cage biomass (*n* = 4 for each experimental group). Samples of liver and epaxial white muscle of three fish per cage were obtained, frozen in liquid nitrogen and maintained at −80 °C until further analysis (*n* = 12 for each experimental group or condition). FCR was also calculated.

#### 4.1.2. Gilthead Sea Bream Juveniles

Two hundred and eighty juveniles (initial body weight 16.3 ± 0.15 g) obtained from the same hatchery were distributed into eight tanks (200 L) (35 fish/tank), as previously described [12]. The reared conditions were the same before indicated. Fish were fed ad libitum (near 5% body weight/day) three times a day (7:30, 14:00 and 21:40) with a commercial diet (D-2 Optibream AE-1P, Skretting, Burgos, Spain). After three weeks of adaptation, four tanks were randomly selected for each experimental group; Control or rBGH. All fish were anesthetized, weighed and sized and intraperitoneally injected with a single dose (6 mg/g body weight) of rBGH or with the same volume of sesame oil for the controls, and recovered into their corresponding tank. At 9 weeks post-injection, body length and weight of all fish were recorded to calculate SGR and CF of each fish. FCR was also calculated. Three fish from each tank (*n* = 12 per condition) were anesthetized and sacrificed and then liver and mesenteric fat weighed in order to calculate HSI and MFI, respectively. Samples of liver, red and white muscle were obtained, immediately frozen in liquid nitrogen and stored at −80°C until further analysis.

The experiments complied with the guidelines of the Council of the European Union (EU 2010/63) and the University of Barcelona (Spain) ethical standards for the use of laboratory animals (CEEA 208/14 and DAAM 7956).

### 4.2. Proximal and Isotopic Composition (δ^15^N and δ^13^C) of Tissues

According to our previous studies [37], samples of liver, red and white muscle were ground in liquid N_2_ using a pestle and mortar to obtain a fine powder. Aliquots of each sample were taken for use in isotopic analyses and to assess the lipid, protein, glycogen and water content. The latter was determined gravimetrically after drying the samples at 95 °C for 24 h. Lipids were extracted as described by Bligh and Dyer [62]; lipid extracts were dried under a N_2_ atmosphere, and total lipids determined gravimetrically. Proteins were purified from defatted tissue samples via precipitation with 10% (*v*/*v*) trifluoroacetic acid. The extracts were dried using a vacuum system (Speed Vac Plus AR, Savant Speed Vac Systems, South San Francisco, CA, USA), and the protein content was calculated from the total N content obtained by elemental analysis (Elemental Analyser Flash 1112, ThermoFinnigan, Bremen, Germany), assuming 1 g of N for every 6.25 g of protein. Glycogen was extracted and purified following alkaline hydrolysis of tissues by boiling with 30% KOH and an alcoholic precipitation, as described by Good et al. [63]. Glycogen content was then assessed using the anthrone colorimetric method described by Fraga [64].

Samples of diet and white muscle were lyophilized and grounded into a homogenous powder for isotopic analysis. Aliquots of the diet and their purified fractions (lipid and protein) and of white muscle, together with their purified tissue fractions (glycogen, lipid and protein), which ranged from 0.3 to 0.6 mg, were weighed in small tin capsules. Samples were analysed to determine the carbon and nitrogen isotope composition using a Mat Delta C Isotope Ratio mass spectrometer (Finnigan MAT, Bremen, Germany) coupled to a Flash 1112 Elemental Analyzer. Isotope ratios (^15^N/^14^N, ^13^C/^12^C) determined by isotope ratio mass spectrometry are expressed in delta (δ) units (parts per thousand, ‰), as follows:δ= [(Rsa/Rst) − 1] × 1000
where Rsa is the ^15^N/^14^N or ^13^C/^12^C ratio of samples and Rst is the ^15^N/^14^N or ^13^C/^12^C ratio of the international standards (Vienna Pee Dee Belemnite for C and air for N). The same reference material analysed over the experimental period was measured with ±0.2‰ precision. Nitrogen and carbon isotopic fractionation values (Δδ^15^N and Δδ^13^C) were calculated as the difference between the δ value in the tissue and their corresponding δ value in the diet.

### 4.3. Nucleic Acid Quantification and Enzyme Activity

White muscle nucleic acid levels (RNA and DNA) were determined using the UV-based procedures for fish samples described by Buckley and Bulow [65]. RNA and DNA from muscle samples were hydrolysed to nucleotides, and their concentrations were calculated based on their absorbance at 260 nm. Nucleic acid concentrations were expressed as μg of RNA or DNA per mg of wet tissue. An aliquot of supernatant was also used to determine protein content [66].

Metabolic enzyme activity was assayed from red and white muscle (CS and COX) or liver (CS, COX, LDH, ALAT, ASAT and HOAD) extracts obtained by homogenizing frozen tissue in detergent solution (1.24 mM TRITON X-100, 1 mM EDTA and 1 mM NaHCO3) and stabilizing solution (3.7 mM EDTA and 5 mM 2-β-mercaptoethanol), 1:1 *v*/*v*, as previously explained [49]. Homogenates were centrifuged at 700 g at 4 °C for 10 min. The microtitration assays (final volume 200 μL) to obtain maximal enzyme activity were performed as follows: ASAT (aspartate aminotransferase, EC 2.6.1.1): 50 mM Tris–HCl buffer (pH 7.4), 10 mM α-ketoglutarate, 0.3 mM NADH, 25 mM l-aspartate (substrate); ALAT (alanine aminotransferase, EC 2.6.1.2): 50 mM Tris–HCl buffer (pH 7.4), 10 mM α-ketoglutarate, 0.3 mM NADH, 25 mM l-alanine (substrate); LDH (lactate dehydrogenase, EC 1.1.2.4): 50 mM Tris–HCl buffer (pH 7.4), 0.16 mM NADH, 1 mM pyruvate (substrate). HOAD (3-hydroxyacyl CoA dehydrogenase, E.C 1.1.1.35): 71.4 mM imidazole buffer (pH 8.0), 2 mM NADH, 2 mM acetoacetyl-CoA (substrate). Citrate synthase (CS, EC 2.3.3.1) activity was determined from absorbance increases at 412 nm of DTNB reagent, using oxaloacetic acid as the substrate, following the method described by Srere [67]. Cytochrome-c-oxidase (COX, EC 1.9.3.1) activity was determined by adapting a commercial kit (CYTOC-OX1, Sigma-Aldrich Inc., St. Louis, MO, USA). This colorimetric assay measures the reduction in ferrocytochrome c absorbance caused by oxidation of the latter by COX.

Redox enzyme activities (CAT, SOD, GR and GPX) were assayed from liver extracts obtained by homogenizing in 1:9 (*w*/*v*) cold buffer (Tris-HCl 0.1M, EDTA 0.1mM, TRITON X-100, pH 7.8), following the previously published procedures [68]. Homogenates were centrifuged at 25,000 g at 4 °C for 30 min. Superoxide dismutase (SOD, EC 1.15.1.1) was determined using method of McCord and Fridovich [69] and adapted for fish [70]: 50 mM potassium phosphate buffer (pH 7.8), 0.1 mM EDTA, 1 mM cytochrome C, 1 mM Xanthine, 0.5 IU/mL xanthine oxidase and sodium hydrosulphite; Catalase (CAT, EC 1.11.1.6) was analysed according to Aebi [71]: 50 mM potassium phosphate buffer, pH 7.0, 10.6 mM H_2_O_2_. Glutathione peroxidase (GPX; EC 1.11.1.9) was assayed by measuring the oxidation of NADPH at 340 nm [72]: 50 mM potassium phosphate buffer, pH 7.2, 1 mM EDTA, 2 mM sodium azide, 0.5-1 U/mL glutathione reductase, 2 mM reduced gluthatione, 0.1 mM NADPH. Glutathione reductase (GR, EC 1.8.1.7) was measured by analysing NADPH oxidation at 340 nm [73]: 0.1 mM potassium phosphate buffer, pH 7.5, 1 mM EDTA, 0.63 mM NADPH and 3.25 mM oxidized glutathione.

All enzymatic activity measurements were performed in 96-well microplates in duplicate at room temperature (20 °C) with a Tecan M200 spectrophotometer (Tecan Trading AG, Männedorf, Switzerland). All reagents, substrates, coenzymes and purified enzymes were purchased from Sigma-Aldrich (Tres Cantos, Spain) and Bio-Rad Laboratories, Inc. (Berkeley, CA, USA). All enzyme activities were expressed in milliunits (mUI) per mg of wet tissue, one unit being the amount of enzyme that converts 1 μmol of substrate per min. Another aliquot of the supernatant was used to determine the protein content in accordance with the Bradford method [65].

### 4.4. Western Blot

Protein was extracted from 100 mg of skeletal white muscle in 1 mL of RIPA buffer supplemented with phosphatase (PMSF and NA_3_VO_4_) and protease inhibitors (P8340, Sigma-Aldrich) using the Precellys^®^ Evolution coupled to a Cryolys cooling system (Bertin Technologies, Montigny-le-Bretonneux, France). Soluble protein concentration was determined by the Bradford’s method using BSA (Sigma-Aldrich, St. Louis, MO, USA) for the standard curve.

Twenty µg of the soluble protein fraction were prepared in a loading buffer (containing SDS and β-mercaptoethanol), heated at 95 °C for 5 min and separated in polyacrylamide gel (12% for CS and PGC1α; 15% for COX4). Following, the proteins were transferred overnight to Immobilon^®^ PVDF-FL 0.2 μm Transfer Membranes (Merck Millipore Ltd., Tullagreen, Cork, Ireland), previously activated in methanol. Total transferred protein was determined by 5 min incubation with Ponceau S (Sigma-Aldrich). The membranes were blocked in 5% skimmed powder milk solution in TBST buffer for 1 h at room temperature and then incubated overnight at 4 °C and in agitation with the corresponding diluted primary antibody. The primary antibodies used were purchased from ABCAM (Cambridge, UK) as follows: rabbit polyclonal anti-Cox IV antibody (ab16056) 1/1000, rabbit polyclonal anti-citrate synthase antibody (ab96600) 1/2000, rabbit polyclonal anti-PGC1α antibody (ab54481) 1/1000. The cross-reactivity of these antibodies with gilthead sea bream was confirmed by the molecular weight of the bands. Subsequently after washing with TBS-T, the membranes were incubated with a secondary antibody: mouse anti-rabbit IgG-HRP (sc-2357) 1/15000. After incubation, membranes were washed with TBS-T, and the immunoreactive bands were developed by an enhanced chemiluminescence kit (Pierce ECL Western Blot Substrate, Thermo Scientific, Alcobendas, Spain). The densitometry of the total transferred protein and the immunoreactive bands was measured using the Quantity One tm software (Bio-Rad), and the relative densitometry was determined for each well.

### 4.5. Proteome Analysis

#### 4.5.1. Protein Extraction and 2-Dimesional Electrophoresis Separation

Approximately 0.3 g of frozen muscle tissue was mechanically powdered in a mortar cooled with liquid nitrogen and homogenized in 3 mL ice-cold phosphate buffer (50 mM, pH 7.0) containing Protease Inhibitor Cocktail (Sigma-Aldrich, St. Louis, MO, USA). An aliquot of 0.3 mL of each sample was stored for bulk protein determination. Homogenates were centrifuged at 15,000 g for 45 min at 4 °C, and supernatants (soluble fraction) were collected. This soluble fraction consists mainly of sarcoplasmic proteins that are easily solubilized at low ionic strength. The protein content of supernatants was determined using the Bio-Rad Protein Assay (Bio-Rad Laboratories), and samples were aliquoted and stored at −80 °C until required. Further protein purification was performed by precipitating the samples in 4 volumes of ice-cold acetone.

According to the previous protocol [36], 300 μg of purified protein was dissolved into 450 μL of rehydration solution containing 7 M urea, 2 M thiourea, 2% *w*/*v* CHAPS, and 0.5% *v*/*v* IPG buffer pH 3−10 NL (Amersham Biosciences Europe, now GE Healthcare, Madrid, Spain), 80 mM DTT and 0.002% of bromophenol blue. The solution was then loaded onto 24-cm, pH 3−10 NL IPG strips. Isoelectric focusing was performed using an IPGphor instrument (Amersham Biosciences), following the manufacturer’s instructions (active rehydration at 50 V for 12 h followed by a linear gradient from 500 to 8000 V until 48,000 V/h) (see more details about gels in Appendix A. The resolved proteins were fixed for 1 h in 40% *v*/*v* methanol containing 10% *v*/*v* acetic acid and stained overnight using colloidal Coomassie Blue G-250.

#### 4.5.2. Gel Image Analysis

Coomassie blue gels were scanned in a calibrated Image Scanner III densitometer (GE Healthcare, Barcelona, Spain) and digital images captured at a resolution of 300 dpi in grey scale mode by Labscan 6.0 software (GE Healthcare, Barcelona, Spain) and saved as uncompressed TIFF files. Gels from six independent biological replicates were analysed using the software package Image Master 2D version 6.01 (GE Healthcare, Barcelona, Spain). Protein spots that varied in abundance between Control and rBGH samples were analysed for significance by unpaired sample *t*-test (SPSS v.16; Chicago, IL, USA). The Shapiro−Wilk test was previously used to ensure the normal distribution of data, and the equality of variances was determined by statistical Levene’s test. The candidate spots selected for MS analysis presented ≥2-fold change in normalized volume and proteins were manually in-gel digested with trypsin (Sequencing grade modified, Promega) following the procedure shown in [36]. Tryptic peptides were extracted from the gel matrix with 10% formic acid and acetonitrile; the extracts were pooled and dried in a vacuum centrifuge.

#### 4.5.3. LC-MS/MS Analysis and Database Search

The dried-down peptide mixtures were analysed in a nanoAcquity liquid chromatographer (Waters, Mildorf, MA, USA)) coupled to an LTQ-Orbitrap Velos (Thermo Scientific, Bremen, Germany) mass spectrometer. The tryptic digests were resuspended in 1% FA solution, and an aliquot was injected for chromatographic separation (more details in Appendix A). Generated raw data files were collected with Thermo Xcalibur (v.2.2) and used to search against the public database Uniprot Actinopterygii (v.23/3/17). A database containing common laboratory contaminant proteins was added to this database. The software used was Thermo Proteome Discoverer (v.1.4.1.14) with Sequest HT as the search engine. The search parameters applied are shown in Appendix A. All possible protein identifications from analyses that met the above criteria were reported for each gel spot. However, the protein identification with the highest score, discarding contaminants, was selected in the case of redundant protein identifications. The Batch-Genes tool produced by GOEAST (http://omicslab.genetics.ac.cn/GOEAST/2019, access on 11 November 2021) performed enrichment analyses of GO annotation terms for biological processes. The proteomics work was done at the Proteomics Platform of Barcelona Science Park, University of Barcelona, a member of ProteoRed-ISCIII network.

### 4.6. Statistical Analysis

Data are presented as mean ± standard error of the mean (SEM) and were analysed using the software IBM SPSS Statistics V.22. A two-way ANOVA was used with condition (rBGH or control groups) as a fixed factor and tank as a random factor, after checking for normal distribution of data and equality of variances using the Shapiro-Wilk and Levene’s test, respectively. As there were not significant effects of factor tank for any of the variables analysed, a Student’s *t*-test comparison was used, being the sample sizes used to compare between the two groups n=4 (tanks per condition) for growth parameters and FCR and *n* = 12 per condition (three fish per 4 tanks) for all the variables, except for HSI and MFI (*n* = 80 per condition). Differences were considered significant at *p* < 0.05.

## Figures and Tables

**Figure 1 ijms-22-13107-f001:**
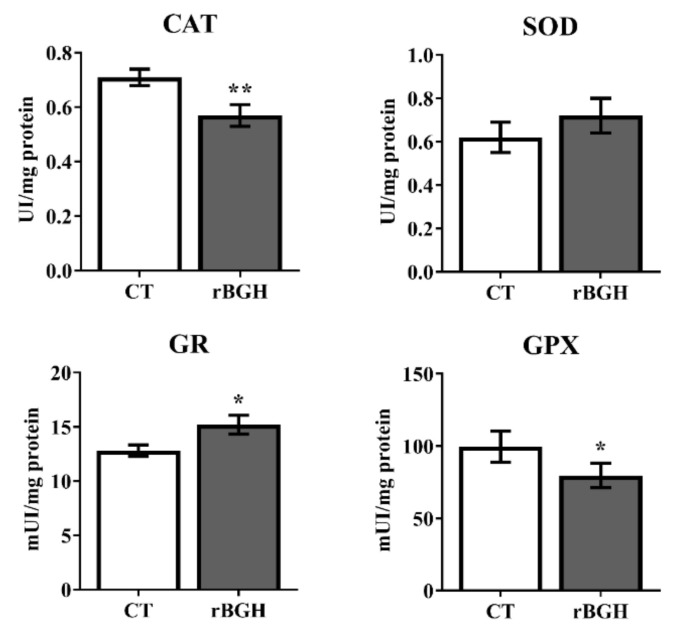
Enzyme activities of redox metabolism in liver after 9 weeks of rBGH injection on gilthead sea bream juveniles. Data are shown as means ± SEM (*n* = 12). Significant differences between control and rBGH groups for each enzyme were determined by Student’s *t*-test and are marked with (*) *p* < 0.05 or (**) *p* < 0.01. Abbreviations: catalase (CAT), superoxide dismutase (SOD), glutathione reductase (GR) and glutathione peroxidase (GPX).

**Figure 2 ijms-22-13107-f002:**
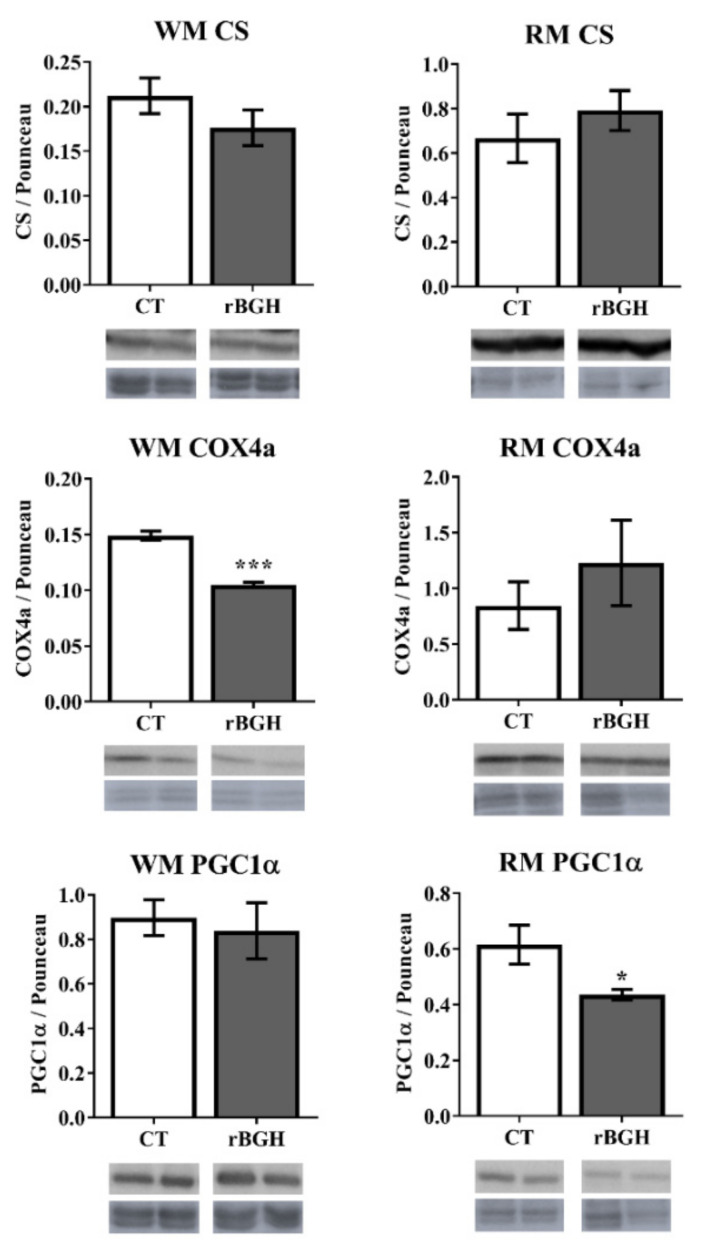
Protein expression of citrate synthase (CS), cytochrome-c-oxidase (COX4a) and peroxisome proliferator-activated receptor gamma coactivator 1-alpha (PGC1α) in white and red muscle after 9 weeks of rBGH injection on gilthead sea bream juveniles. Specific immunoreactive bands were normalized to total protein staining with Ponceau S and shown as means ± SEM (*n* = 6). Intermembrane variability was normalized using a random sample as a protein load control (more details in Appendix A). Significant differences between control and rBGH groups for each protein were determined by Student’s *t*-test and are marked with (*) *p* < 0.05; (***) *p* < 0.001. Abbreviations: WM (white muscle) and RM (red muscle).

**Figure 3 ijms-22-13107-f003:**
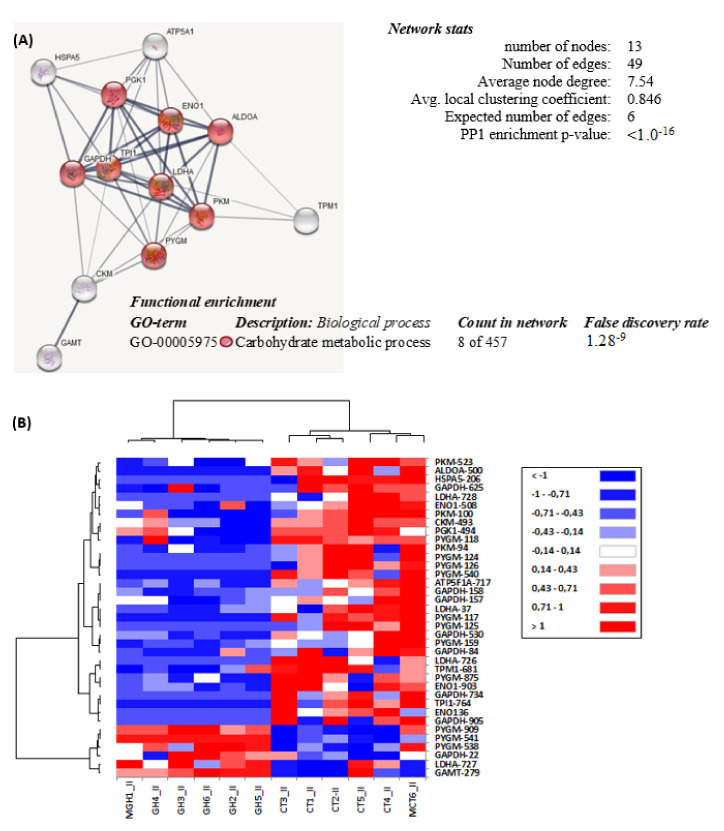
White muscle proteomic profile of differentially expressed proteins between rbGH group and control group of gilthead sea bream. (**A**) The network analysis of protein–protein interactions among proteins grouped into 1 functional category, according to Gene Ontology enrichment analysis by GOEAST. In this network, nodes are proteins, the thickness of the lines indicates the degree of confidence prediction of the interaction, according to the STRING databases. The average of local clustering coefficient was 0.846 for “Carbohydrate metabolic process” (GO: 0005975, *p*-value: 1.0^−^^16^) (see Appendix A). (**B**) Heat map of the hierarchical clustering of the relative abundance of differentially expressed proteins between the white muscle of rbGH group and control group (Perseus program). Every line represents one independent sample of each group, while the different protein spots are represented by individual rows. Red indicates high levels of expression while blue represents low levels. The intensity of the colours represents the relative abundance. Raw data of each spot are included in Appendix A). Protein acronyms correspond to Gene Symbol (see Table 5 for details).

**Table 1 ijms-22-13107-t001:** Somatic growth parameters and feed conversion rate (FCR) after 6 and 9 weeks of rBGH injection on gilthead sea bream fingerlings and juveniles.

	Fingerlings			Juveniles		
	Control	rbGH		Control	rbGH	
Initial B.W.	1.04 ± 0.02	1.01 ± 0.05		16.3 ± 0.06	16.3 ± 0.34	
Final B.W.	8.07 ± 0.25	8.84 ± 0.18	*	57.2 ± 1.49	70.6 ± 2.20	**
SGR	4.9 ± 0.08	5.2 ± 0.07	*	1.98 ± 0.02	2.33 ± 0.02	***
CF	1.5 ± 0.01	1.5 ± 0.01		2.53 ± 0.03	2.39 ± 0.04	
HSI	1.6 ± 0.04	1.4 ± 0.04	**	1.5 ± 0.06	1.2 ± 0.05	**
MSI	-	-		34.0 ± 0.63	34.8 ± 0.34	
MFI	1.5 ± 0.07	1.3 ± 0.07		1.5 ± 0.06	1.2 ± 0.06	
FCR	1.7 ± 0.13	1.6 ± 0.12		1.7± 0.14	1.3 ± 0.05	*

Mean ± SEM. N = 4 for SGR, CF and FCR; N = 80 for HSI and MFI; N = 12 for MSI. BW: body weight. SGR: specific growth rate [100·(ln final BW-ln initial BW)]·days^−1^. Condition Factor = 100· BW/TL^3.^ HSI: Hepatosomatic Index = [g liver·100g B.W.^−1^]. MSI: Muscle-somatic Index [g muscle·100g B.W.^−1^]. MFI: Mesenteric fat Index [g fat·100g B.W.^−1^]. FCR: Feed conversion rate = [feed eaten·weight gain^−1^]. Significant differences by Student’s *t*-test: (*) *p* < 0.05; (**) *p* < 0.01; (***) *p* < 0.001.

**Table 2 ijms-22-13107-t002:** Proximal composition, RNA and DNA contents and CS and COX activities in white muscle after 6 or 9 weeks of rBGH injection on gilthead sea bream fingerlings and juveniles, respectively.

	Fingerlings			Juveniles		
	Control	rbGH		Control	rbGH	
Composition						
Protein (% w.w.)	19.2 ± 0.17	19.2 ± 0.18		19.8 ± 0.28	19.7 ± 0.22	
Lipids (% w.w.)	2.2 ± 0.12	1.5 ± 0.08	***	2.4 ± 0.12	1.5 ± 0.10	***
Glycogen (% w.w.)	0.15 ± 0.01	0.17 ± 0.02		0.25 ± 0.04	0.46 ± 0.05	**
Wet weight (%)	78.5 ± 0.14	78.5 ± 0.14		75.1 ± 0.26	76.5 ± 0.22	*
RNA (µg/mg prot)	6.7 ± 0.19	7.3 ± 0.56		4.1 ± 0.09	5.4 ± 0.20	***
DNA (µg/mg prot)	1.46 ± 0.07	1.42 ± 0.13		0.9 ± 0.03	1.0 ± 0.05	
RNA/DNA	4.7 ± 0.27	4.8 ± 0.28		4.7 ± 0.18	5.7 ± 0.32	**
Enzyme activities						
CS ^1^	77.7 ± 3.50	81.0 ± 5.07		42.3 ± 1.98	44.7 ± 1.81	
COX ^1^	27.5 ± 1.25	26.9 ± 2.19		15.1 ± 0.56	14.5 ± 0.83	
COX/CS	0.35 ± 0.02	0.34 ± 0.03		0.37 ± 0.03	0.33 ± 0.02	

Mean ± SEM. N = 10 for fingerlings and N = 12 for juveniles. ^1^ mUI/mg prot. CS: citrate synthase; COX: cytochrome-c-oxidase. Significant differences by Student’s *t*-test: (*) *p* < 0.05; (**) *p* < 0.01; (***) *p* < 0.001.

**Table 3 ijms-22-13107-t003:** Proximal composition and metabolic enzyme activities in red muscle and liver after 9 weeks of rBGH injection on gilthead sea bream juveniles.

	Juveniles					
	Red Muscle			Liver		
	Control	rbGH		Control	rbGH	
Composition						
Protein (% w.w.)	15.7 ± 1.29	16.9 ± 1.49		12.5 ± 0.62	12.5 ± 0.79	
Glycogen (% w.w.)	0.45 ± 0.06	0.75 ± 0.02	**	12.5 ± 0.85	9.9 ± 0.51	*
Lipids (% w.w.)	19.2 ± 0.87	14.4 ± 1.70		16.8 ± 0.77	13.6 ± 0.97	*
Enzyme activities						
CS ^1^	467 ± 23.1	515 ± 51.2		34.3 ± 1.76	40.1 ± 1.37	*
COX ^1^	131 ± 10.3	125 ± 9.92		121.8 ± 3.8	114.4 ± 5.20	
HOAD ^2^				5.09 ± 0.35	5.14 ± 0.20	
LDH ^1^				8.01 ± 0.68	8.07 ± 0.41	
ALAT ^1^				45.6 ± 1.22	39.4 ± 1.01	**
ASAT ^1^				77.6 ± 4.53	81.3 ± 3.16	

Mean ± SEM. N = 12. Significant differences by Student’s *t*-test: (*) *p* < 0.05; (**) *p* < 0.01. ^1^ mUI/mg protein. ^2^ UI/mg protein. CS: citrate synthase; COX: cytochrome-c-oxidase; LDH: lactate dehydrogenase; ALAT: alanine transaminase; ASAT: Aspartate transaminase.

**Table 4 ijms-22-13107-t004:** Isotopic composition (^15^N/^13^C) in white muscle after 6 or 9 weeks of rBGH injection on gilthead sea bream fingerlings and juveniles.

	Fingerlings			Juveniles		
	Control	rbGH		Control	rbGH	
δ^13^ C-muscle	−20.60 ± 0.03	−20.45 ± 0.03	**	−20.49 ± 0.12	−19.77 ± 0.06	***
δ^13^ C-lipid	−26.15 ± 0.05	−25.97 ± 0.06	*	−25.61 ± 0.02	−25.46 ± 0.03	***
δ^13^ C-glycogen	−20.91 ± 0.14	−20.60 ± 0.14		−19.93 ± 0.19	−19.41 ± 0.17	*
δ^13^ C-protein	−21.51 ± 0.05	−21.89 ± 0.36		−20.49 ± 0.02	−20.56 ± 0.05	
δ^15^ N-muscle	12.19 ± 0.04	12.13 ± 0.06		9.64 ± 0.11	9.24 ± 0.06	**
δ^15^ N-protein	13.47 ± 0.06	13.19 ± 0.09	*	10.59 ± 0.10	10.22 ± 0.07	**
∆^15^ N-muscle ^1^	2.57 ± 0.04	2.51 ± 0.07		0.02 ± 0.11	-0.38 ± 0.06	**
∆^15^ N-protein ^1^	3.73 ± 0.05	3.45 ± 0.09	*	0.85 ± 0.10	0.48 ± 0.07	*

Mean ± SEM. N = 10 for fingerlings and N = 12 for juveniles. Significant differences by Student’s *t*-test: (*) *p* < 0.05; (**) *p* < 0.01; (***) *p* < 0.001. ^1^ 15N Fractionation = δ^15^N (in muscle or protein fraction)—δ^15^N diet. Diet: δ^15^N is 9.62 ± 0.05 in the whole diet and 9.74 ± 0.05 in the dietary protein fraction.

**Table 5 ijms-22-13107-t005:** Identification of the 37 differentially expressed proteins of white muscle after 9 weeks of rBGH injection on gilthead sea bream juveniles.

^a^ SPOT	Accession No.	^b^ Protein Name	Species		^c^ Symbol	Theorical KDa/pI	Observed Kda/pI	^d^ score	^e^ Peptides (Unique)	^f^ SC (%)	^g^ FC	^h^ *p*-Value	UniprotKB
**Cellular Metabolic Process: GO: 0044237**													
**Carbohydrate Metabolic Process: GO: 0005975** (*p*-Value: 2.3^−^^47^)													
125	I3JBN0	Alpha-1,4 glucan phosphorylase	*Oreochromis niloticus*		PYGM	97.1/7.1	95.0/8.5	233.68	28 (2)	34.68	0.24	0.0102	P11217
538	I3JBN0	Alpha-1,4 glucan phosphorylase	*Oreochromis niloticus*		PYGM	97.1/7.1	96.0/6.9	991.63	35 (6)	40.50	1.39	0.0422	P11217
540	I3JBN0	Alpha-1,4 glucan phosphorylase	*Oreochromis niloticus*		PYGM	97.1/7.1	96.0/6.9	991.11	33 (5)	39.07	0.74	0.0026	P11217
117	Q4SFP9	Alpha-1,4 glucan phosphorylase (Fragment)	*Tetraodon nigroviridis*		PYGM	97.2/6.9	95.0/7.3	1085.21	36 (1)	41.74	0.40	0.0003	P11217
541	G3QBP8	Alpha-1,4 glucan phosphorylase	*Gasterosteus aculeatus*		PYGM	83.5/6.9	97.0/6.8	100.04	7 (2)	13.36	1.51	0.0001	P11217
909	Q4SFP9	Alpha-1,4 glucan phosphorylase (Fragment)	*Tetraodon nigroviridis*		PYGM	97.2/6.9	95.0/7.2	244.66	30 (2)	37.57	3.21	0.0018	P11217
124	I3JBN0	Alpha-1,4 glucan phosphorylase	*Oreochromis niloticus*		PYGM	97.1/7.1	96.0/8.3	524.38	36 (3)	40.26	0.32	0.0117	P11217
118	A0A147AQX3	Alpha-1,4 glucan phosphorylase	*Fundulus heteroclitus*		PYGM	102.7/7.3	97.0/7.4	907.09	33 (3)	37.95	0.34	0.0285	P11217
126	A0A147AQX3	Alpha-1,4 glucan phosphorylase	*Fundulus heteroclitus*		PYGM	102.7/7.3	96.0/8.6	304.63	31 (4)	33.22	0.45	0.0275	P11217
875	I3JBN0	Alpha-1,4 glucan phosphorylase	*Oreochromis niloticus*		PYGM	97.1/7.1	109.0/6.8	309.30	32 (3)	38.36	0.23	0.0211	P11217
494	I3KL67	Phosphoglycerate kinase	*Oreochromis niloticus*		PGK1	44.5/6.9	50.0/6.9	905.26	20 (3)	48.68	0.70	0.0244	P00558
157	Q155W8	Glyceraldehyde-3-phosphate dehydrogenase	*Sparus aurata*		GAPDH	36.0/8.4	20.2/8.2	705.47	19 (6)	72.67	0.46	0.0341	P04406
158	Q155W8	Glyceraldehyde-3-phosphate dehydrogenase	*Sparus aurata*		GAPDH	36.0/8.4	24.9/7.8	463.46	15 (7)	69.07	0.29	0.0548	P04406
84	Q155W8	Glyceraldehyde-3-phosphate dehydrogenase	*Sparus aurata*		GAPDH	36.0/8.4	67.0/8.2	1083.31	15 (7)	67.27	0.34	0.0289	P04406
625	Q155W8	Glyceraldehyde-3-phosphate dehydrogenase	*Sparus aurata*		GAPDH	36.0/8.4	25.3/8.0	401.04	11 (11)	60.96	0.53	0.0227	P04406
905	Q155W8	Glyceraldehyde-3-phosphate dehydrogenase	*Sparus aurata*		GAPDH	36.0/8.4	70.0/7.9	33.25	6 (6)	32.73	0.18	0.0368	P04406
22	Q155W8	Glyceraldehyde-3-phosphate dehydrogenase	*Sparus aurata*		GAPDH	36.0/8.4	44.0/6.8	126.74	15 (6)	57.06	2.40	0.0285	P04406
530	Q155W8	Glyceraldehyde-3-phosphate dehydrogenase	*Sparus aurata*		GAPDH	36.0/8.4	87.0/8.1	100.59	10 (5)	49.25	0.14	0.0293	P04406
734	Q155W8	Glyceraldehyde-3-phosphate dehydrogenase	*Sparus aurata*		GAPDH	36.0/8.4	108/8.2	83.44	12 (7)	62.46	0.65	0.0081	P04406
37	O13276	L-lactate dehydrogenase A chain	*Sphyraena argentea*		LDHA	36.4/8.0	69.0/6.9	942.71	17 (2)	37.95	0.37	0.0176	P00338
728	O13276	L-lactate dehydrogenase A chain	*Sphyraena argentea*		LDHA	36.4/8.0	44.0/6.9	557.60	12 (3)	32.23	0.50	0.0348	P00338
726	O13276	L-lactate dehydrogenase A chain	*Sphyraena argentea*		LDHA	36.4/8.0	41.0/6.9	868.73	14 (1)	34.34	0.40	0.0094	P00338
727	O13276	L-lactate dehydrogenase A chain	*Sphyraena argentea*		LDHA	36.4/8.0	48.0/6.9	559.10	14 (4)	34.34	1.92	0.0279	P00338
508	A0A147AE36	Alpha-enolase	*Fundulus heteroclitus*		ENO1	47.5/6.8	64.0/7.3	1028.05	20 (1)	58.29	0.55	0.0220	P06733
903	A0A147AE36	Alpha-enolase	*Fundulus heteroclitus*		ENO1	47.5/6.8	65.0/7.7	865.01	15 (1)	47.24	0.40	0.0322	P06733
36	A0A146XBN2	Alpha-enolase	*Fundulus heteroclitus*		ENO1	46.4/6.7	65.0/6.6	901.83	15 (1)	50.70	0.53	0.0086	P06733
94	A0A0F8AK35	Pyruvate kinase	*Larimichthys crocea*		PKM	58.2/7.7	73.0/8.5	819.93	15 (4)	36.04	0.39	0.0216	P14618
523	Q8QGU8	Pyruvate kinase	*Takifugu rubripes*		PKM	58.0/7.9	74.0/7.6	657.85	18 (2)	34.53	0.44	0.0037	P14618
100	A0A0F8AK35	Pyruvate kinase	*Larimichthys crocea*		PKM	58.2/7.7	74.0/8.0	195.81	8 (1)	21.51	0.22	0.0120	P14618
500	H2TGY6	Fructose-bisphosphate aldolase	*Takifugu rubripes*		ALDOA	39.6/8.3	54.0/7.7	283.00	12 (1)	40.22	0.56	0.0051	P04075
764	A0A1A8A8E2	Triosephosphate isomerase	*Nothobranchius furzeri*		TPI1	26.5/7.3	29.0/7.3	69.63	7 (7)	36.84	0.29	0.0010	P60174
**Others**													
279	Q71N41	Guanidinoacetate N-methyltransferase	*Danio rerio*		GAMT	26.7/6.3	33.0/6.0	513.76	5 (2)	28.21	1.50	0.0138	Q14353
493	A0A146WHL4	Creatine kinase M-type	*Fundulus heteroclitus*		CKM	42.8/6.8	53.0/6.9	1117.72	15 (1)	41.10	0.46	0.0123	P06732
206	A0A0F8AHC2	Glucose-regulated protein	*Larimichthys crocea*		HSPA5	82.3/5.6	95.0/5.6	152.41	19 (3)	29.88	0.64	0.0088	P11021
717	A0A0F6MX10	ATP synthase subunit alpha	*Sparus aurata*		ATP5F1A	59.6/9.1	68.0/6.9	94.37	15 (15)	38.66	0.15	0.0185	P25705
681	D6PVP3	Tropomyosin	*Epinephelus coioides*		TPM1	32.7/4.7	45.0/4.7	183.97	17 (2)	46.13	0.34	0.0080	P09493

^a^ Spot number. ^b^ Protein name derived from BLASTp sequence analysis. ^c^ Symbol of gene product from GeneCards v3.07. ^d^ MASCOT score obtained > score corresponding to *p* < 0.05 (probability 95%). ^e^ Number of total peptides observed and in parentheses unique peptides from the LC-MS/MS analysis. ^f^ Coverage percentage of the peptide sequence homology. ^g^ Intensity fold between the two conditions (rBGH/control: >0, up-regulated; <0, down-regulated). ^h^ *t*-test (*n* = 6).

## Data Availability

The datasets used and/or analyzed during the current study are available from the corresponding author upon reasonable request.

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
