# Peer review of "Recombinant Bovine Growth Hormone-Induced Metabolic Remodelling Enhances Growth of Gilthead Sea-Bream (Sparus aurata): Insights from Stable Isotopes Composition and Proteomics"

_ijms, 2021, doi:10.3390/ijms222313107_

Round 1

Reviewer 1 Report

In the reported study, Blasco et al. continued their elucidation of the effect of bovine growth hormone on the growth of gilthead seabream fingerlings and juveniles 6- and 9-weeks post-treatment, respectively. The experiments were appropriately designed, methods were sound, data presentation and analyses were fitting, and the discussion was thorough. The authors may add a justification regarding the duration post-treatment (6 weeks and 9 weeks) when samples were collected? Was there any previous study showing until when the rBGH is effective? Were there any mortalities observed throughout the study? In the Discussion, can authors comment about the feasibility of injecting rBGH in individual fish at the farm level? For quicker reference for readers, the authors may consider adding in the figure legends what the abbreviations from each graph represent.

Author Response

Response to Reviewer 1 Comments

The authors may add a justification regarding the duration post-treatment (6 weeks and 9 weeks) when samples were collected? Was there any previous study showing until when the rBGH is effective?

Previous studies in other species revealed that the rBGH treatment reduces the plasma levels of endogenous GH in coho salmon even after 14 weeks post-injection (Raven et al. (2012), suggesting that the rBGH treatment still was in circulation, and therefore effective, by that time. In fact, Leedom et al. (2002) using a dose of rBGH four to six times lower than that in our studies (1 mg/g B.W.) found that after 70 days of treatment in tilapia, the rBGH plasma levels were very high (around 200 ng/ml), more than 10-fold higher than the normal GH levels. Moreover, the authors reported that rBGH remain at detectable levels up to 140 days post-injection in coho salmon (McLean et al., 1997). According to this literature, we anticipated that 6 and 9 weeks post-treatment were two time points in which circulating rBGH levels would remain high enough to remain effective, and that would be a sufficient long-term treatment to detect accumulated differences between groups. Later, our own results obtained with the GH  gene expression levels from the pituitary of juveniles fish injected with rBGH, demonstrated that, indeed, the levels of circulating rBGH at 12 weeks post-treatment in gilthead sea bream remained high enough to induce a downregulation of the expression of endogenous GH (Velez et al. 2019), confirming that 6 and 9 weeks was a good choice. Now, an additional comment supporting this has been added in lines 204-205 of the discussion.

Were there any mortalities observed throughout the study?

We did not observe any mortalities along the study.

In the Discussion, can authors comment about the feasibility of injecting rBGH in individual fish at the farm level?

The administration of rBGH has been successful increasing feed efficiency, protein deposition into the carcass and milk production in beef cattle (Dalke et al., 1992; Dohoo et al., 2003). This issue is surrounded by great controversy due to associated problems in animals (i.,e., mastitis), as well as possible effects on the consumer (for example, milk with higher levels of growth factors). Although the research in fish shows that rBGH is a good inductor of growth, the current legislations and the consumers’ criteria are not in line with the idea of using rBGH in fish. The purpose of this study was never to apply this to fish, but to know their potential to grow to later find apply approaches or natural conditions where the secretion of IGF-I/GH is enhanced. These two last sentences have been also added in the text for clarification (lines 95-99).

For quicker reference for readers, the authors may consider adding in the figure legends what the abbreviations from each graph represent.

Now the abbreviations within each graph have been added in the figure legends (see figure legends of figure 1 and 2).

Reviewer 2 Report

The manuscript ijms-1483986 reports experimental data on the effect of GH on proteins and lipids usage, monitoring liver and muscles of the gilthead sea bream.

The topic is extremely interesting, the results are new and increase the knowledge of fish physiology, opening also the way for some specific applicative studies in the field of aquaculture. The methodological approach is correct and adequate to the research.

The manuscript is well organized and well written. I found only a few typos in the text, easily solved with a quick revision. The discussion is very well argued and needs only a little implementation.

In particular, the authors should better discuss the variations in the activities of antioxidant enzymes. In fact, the decrease in GPX activity could suggest a decrease in the formation of H2O2, confirmed by the decrease in CAT. This is strange because the implementation of lipid metabolism favors the formation of ROS. However, there is an increase in the activity of GR, an enzyme that usually complementarity works with GPX. A possible interpretation is that either the GSH is directly involved in the reduction of ROS produced by the increased lipid metabolism (and therefore, when GSSG is formed, the intervention of the GR is required for its reduction in GSH) or the GSH works as a reducing factor of other antioxidant enzymes such as peroxiredoxins. These enzymes are receiving particular attention from the scientific community as they seem to represent an important line of defense against the increase in the rate of H2O2 formation. Moreover, this has also been observed in fish that use lipids as a primary energy, such as Antarctic fish (Tolomeo et al., 2016, 2019)

Tolomeo A.M., Carraro A., Bakiu R., Toppo S., Place S.P., Ferro D., Santovito G. (2016) Peroxiredoxin 6 from the Antarctic emerald rockcod: molecular characterization of its response to warming. J. Comp. Physiol. B 186, 59-71. doi: 10.1007/s00360-015-0935-3

Tolomeo A.M., Carraro A., Bakiu R., Toppo S., Garofalo F., Pellegrino D., Gerdol M., Ferro D., Place S.P., Santovito G. (2019) Molecular characterization of novel mitochondrial peroxiredoxins from the Antarctic emerald rockcod and their gene expression in response to environmental warming. Comp. Biochem. Physiol. C 255, 108580. doi: 10.1016/j.cbpc.2019.108580

I consider that this manuscript is appropriate for publication in International Journal of Molecular Sciences, and I suggest to accept it after minor revision.

Author Response

Response to Reviewer 2 Comments

The manuscript is well organized and well written. I found only a few typos in the text, easily solved with a quick revision.

We have revised and corrected the typos in the MS.

The discussion is very well argued and needs only a little implementation.

In particular, the authors should better discuss the variations in the activities of antioxidant enzymes. In fact, the decrease in GPX activity could suggest a decrease in the formation of H2O2, confirmed by the decrease in CAT. This is strange because the implementation of lipid metabolism favors the formation of ROS. However, there is an increase in the activity of GR, an enzyme that usually complementarity works with GPX. A possible interpretation is that either the GSH is directly involved in the reduction of ROS produced by the increased lipid metabolism (and therefore, when GSSG is formed, the intervention of the GR is required for its reduction in GSH) or the GSH works as a reducing factor of other antioxidant enzymes such as peroxiredoxins. These enzymes are receiving particular attention from the scientific community as they seem to represent an important line of defense against the increase in the rate of H2O2 formation. Moreover, this has also been observed in fish that use lipids as a primary energy, such as Antarctic fish (Tolomeo et al., 2016, 2019)

We thank the referee for their recommendation regarding better explanation of antioxidant enzyme variations, as well as their suggestion about peroxyredoxins that might be involved. Thus, one of the suggested references has been added in the text (Tolomeo et al., 2019) to support this hypothesis (lines 296-304), and the discussion in this point has been improved.